

# A Complete Parameterization of the Relative Humidity and Wavelength Dependence of the Refractive Index of Hygroscopic Inorganic Aerosol Particles

Michael I. Cotterell, Rose E. Willoughby, Bryan R. Bzdek, Andrew J. Orr-Ewing and Jonathan P. Reid

School of Chemistry, University of Bristol, Bristol, BS8 1TS, UK

Correspondence to: Jonathan P. Reid (j.p.reid@bristol.ac.uk)

**Abstract**. Calculations of aerosol radiative forcing require knowledge of wavelength-dependent aerosol optical properties, such as single scattering albedo. These aerosol optical properties can be calculated using Mie theory from knowledge of the key microphysical properties of particle size and refractive index, assuming that atmospheric particles are well-approximated

to be spherical and homogeneous. We provide refractive index determinations for aqueous aerosol particles containing the key atmospherically relevant inorganic solutes of $NaCl$, $NaNO_3$, $(NH_4)_2SO_4$, $NH_4HSO_4$ and $Na_2SO_4$, reporting the refractive index variation with both wavelength (400 – 650 nm) and relative humidity (from 100% to the efflorescence value of the salt). The accurate and precise retrieval of refractive index is performed using single particle cavity ring-down spectroscopy. This approach involves probing a single aerosol particle confined in a Bessel laser beam optical trap through a combination

of extinction measurements by cavity ring-down spectroscopy and elastic light scattering measurements. Further, we assess the accuracy of these refractive index measurements, comparing our data with previously reported data sets from different measurement techniques but at a single wavelength. Finally, we provide a Cauchy dispersion model that parameterizes refractive index measurements in terms of both wavelength and relative humidity. Our parameterizations should provide useful information to researchers requiring an accurate and comprehensive treatment of the wavelength and relative humidity

dependence of the inorganic component of atmospheric aerosol.





## 1. Introduction

The current best estimate of aerosol effective radiative forcing (relative to the year 1750) is -0.9 $\left(\begin{smallmatrix}+0.8\\-1.0\end{smallmatrix}\right)$ W m$^{-2}$, where the uncertainty range represents the 5-95% confidence limits (Alexander et al., 2013). Reducing the large uncertainty associated with aerosol radiative forcing (RF) is crucial to improving the representation of aerosol in climate models. Top-of-the-

atmosphere radiative forcing (RF$_{TOA}$) is a common metric for assessing the contribution of different aerosol particles to the warming or cooling of Earth's atmosphere (Alexander et al., 2013; Erlick et al., 2011; Haywood and Boucher, 2000; Ravishankara et al., 2015). RF$_{TOA}$ can be estimated for a uniform, optically thin layer of aerosol in the lower troposphere using (Dinar et al., 2008; Haywood and Shine, 1995; Zarzana et al., 2014):

$$\mathrm{RF}_{TOA} = S_0 D T_{at}^2 (1 - A_c)\left[2R_s(1 - \overline{\omega}) - \overline{\beta}\,\overline{\omega}(1 - R_s)^2\right] \qquad 1$$

in which $S_0$ is the solar constant (1370 Wm$^{-2}$), $D$ is the fractional day length, $T_{at}$ is the solar atmospheric transmittance, $A_c$ is

the fractional cloud cover and $R_s$ is the surface reflectance. The use of satellite, aeroplane and ground-based observations allows quantification of the geographical variation in $D$, $T_{at}$, $A_c$ and $R_s$. Importantly, equation (1) indicates the dependence of RF$_{TOA}$ on the spectrally-weighted single scattering albedo, $\overline{\omega}$, and the spectrally-weighted backscattered fraction, $\overline{\beta}$. Under the assumptions that a particle is both spherical and homogeneous, $\overline{\omega}$ and $\overline{\beta}$ (in addition to other aerosol optical properties) can be calculated using Mie theory with input values of the aerosol particle size ($a$) and the wavelength ($\lambda$) dependent

complex refractive index (RI, $m = n + ki$) (Bohren and Huffman, 1998; Hess et al., 1998; Levoni et al., 1997). Although measurements of particle size and size distribution are relatively straightforward using techniques such as aerodynamic particle sizing and differential mobility analysis (DMA), the measurement of $m$ is more challenging, particularly as aqueous aerosol droplets commonly exist at supersaturated solute concentrations which are not accessible in bulk measurements. Zarzana *et al*. estimated that an uncertainty in the real component of the RI, $n$, of 0.003 (0.2%) leads to an uncertainty in RF

of 1%, for non-absorbing (NH$_4$)$_2$SO$_4$ particles with a 75 or 100 nm radius (Zarzana et al., 2014). Meanwhile, Moise *et al*. reported that variation in $n$ from 1.4 to 1.5 resulted in an increase in the radiative forcing by 12% (Moise et al., 2015). Therefore, any strategy to reduce the uncertainty in RF$_{TOA}$ that starts from characterisations of aerosol microphysical properties (a *bottom-up* approach) requires the development of new *in situ* techniques for the accurate characterisation of wavelength and relative humidity (RH) dependent refractive indices.

Tang *et al*. levitated single, laboratory-generated aerosol particles in an electrodynamic balance, measuring relative changes in the droplet mass and recording the elastic light scattered at a fixed angle relative to the propagation direction of a 633 nm laser beam (Tang et al., 1997; Tang and Munkelwitz, 1994). By comparing the measured light scattering traces to calculations from Mie theory, $n$ was parameterised as a function of RH for a range of inorganic-aqueous solution droplets,

key components of non-absorbing atmospheric aerosol. Tang's parameterisations of $n_{633}$ for aqueous inorganic solutes have become a benchmarking standard for new techniques measuring refractive index, (Cotterell et al., 2015b; Hand and





Kreidenweis, 2002; Mason et al., 2012, 2015) and are used as reference RI data for RF calculations of aqueous inorganic aerosol (Erlick et al., 2011). However, Tang's measurements were limited to $\lambda$ = 633 nm and knowledge of the optical dispersion is required to calculate spectrally weighted values of the single scattering albedo and backscatter function.

Another method for determining the RI of a particle is to fit the measured size dependence of optical cross sections to Mie theory (Moise et al., 2015). In particular, the interaction of light with an aerosol particle is governed by the particle extinction cross section, $\sigma_{ext}$, (Bohren and Huffman, 1998; Miles et al., 2011a) and measurements of $\sigma_{ext}$ using cavity enhanced methods are, in principle, highly precise. Ensemble broadband cavity enhanced spectroscopy (E-BBCES) involves $\sigma_{ext}$ measurements on a cloud of aerosol particles over a broad range of wavelengths. Zhang *et al.* have developed an E-
BBCES instrument for measurements of $\sigma_{ext}$ over the wavelength range 445 – 480 nm and reported measurements of $(NH_4)_2SO_4$ aerosol (Zhao et al., 2013). Rudich and co-workers developed the E-BBCES technique extensively for the wavelength range 360 – 420 nm (Flores et al., 2014a, 2014b; Washenfelder et al., 2013). From E-BBCES measurements on Suwannee River fulvic acid aerosol (a weakly absorbing species), *m* was measured to be 1.71 (± 0.02) + 0.07 (± 0.06)*i* at 360 nm and 1.66 (± 0.02) + 0.06 (±0.04)*i* at 420 nm (Washenfelder et al., 2013). These uncertainties in the retrieved *m* limit the
accuracy of RF calculations.

Ensemble cavity ring-down spectroscopy (E-CRDS) is a related cavity enhanced spectroscopy technique used for $\sigma_{ext}$ measurements for aerosol particles in both the laboratory (Dinar et al., 2008; Lang-Yona et al., 2009; Mason et al., 2012) and in the field (Baynard et al., 2007; Langridge et al., 2011). In E-CRDS, a flow of aerosol is introduced into an optical cavity
consisting of two highly reflective mirrors in which the aerosol ensemble is probed by CRDS and an extinction coefficient, $\alpha_{ext}$, is measured. In combination with measurements of particle number concentration, *N*, using a condensation particle counter, $\sigma_{ext}$ is calculated using $\sigma_{ext} = \alpha_{ext}/N$. However, uncertainty in the population distribution of aerosol within the cavity ring-down beam and errors in the measured *N* can lead to imprecise measurements of $\sigma_{ext}$ (Miles et al., 2011b). By size selecting aerosol using a DMA prior to admitting an ensemble into the optical cavity, the variation in $\sigma_{ext}$ with particle
size is measured and the particle RI can be retrieved. However, the accuracy and precision in the retrieved RIs are often too poor for reliable RF calculations owing to the combination of imprecise $\sigma_{ext}$ measurement and the significant systematic errors that can derive from the DMA size selection process. Mason *et al*. reported a *precision* of ±0.02 (~1.4%) in the retrieved *n* from E-CRDS measurements of NaNO$_3$ particles at a range of RH values (Mason et al., 2012), while Miles *et al*. found that errors in the measured *N* can alone introduce a ~2.5% uncertainty in the retrieved *n,* therefore limiting the
precision of this technique. Under a scenario incorporating best-case errors in variables governing E-CRDS $\sigma_{ext}$ measurements, a theoretical study by Zarzana and co-workers estimated the *accuracy* in the E-CRDS retrieved *n* to be 0.6% at best from simulations of $\sigma_{ext}$ measurements for non-absorbing $(NH_4)_2SO_4$ particles at twelve discrete particle sizes (Zarzana et al., 2014).



Probing a single particle, instead of an aerosol ensemble, resolves many of the problems inherent to E-CRDS that degrade the RI retrieval precision. We have previously reported the application of two single particle cavity ring-down spectrometers (SP-CRDS) in measurement of $\sigma_{ext}$ at wavelengths of $\lambda = 405$ and 532 nm for single particles confined within a Bessel laser beam (BB) optical trap (Cotterell et al., 2015a, 2015b; Mason et al., 2015; Walker et al., 2013). No measurement is required

of particle number density and measurements of $\sigma_{ext}$ are made with continuous variation in the particle size as the particle evolves with time, either through an RH change or through particle-gas partitioning of semi-volatile components. In our measurements, the particle size is precisely determined from fitting the angularly-resolved elastic light scattering distributions to Mie theory. For single component evaporation measurements, we measured the precision in retrievals of $n$ to be ±0.0007 for the 532-nm SP-CRDS instrument (Mason et al., 2015) and ±0.0012 using the 405-nm SP-CRDS instrument

(Cotterell et al., 2015b). Also, we demonstrated the retrieval of RI from hygroscopic response measurements for aqueous droplets containing inorganic solutes (Cotterell et al., 2015a, 2015b). By simulating $\sigma_{ext}$ data using the parameters of the 532-nm SP-CRDS instrument, we demonstrated the expected retrieval accuracy in $n$ to be 0.0002 (0.014%) for single component particles evaporating over the radius range $1 - 2$ μm. Meanwhile, we showed that the retrieval accuracy is < 0.001 for coarse mode particles when $n$ varies with particle size, such as in a hygroscopic response measurement (Cotterell et al., 2016).

In this paper, we report the application of the 405-nm and 532-nm SP-CRDS instruments for the measurement of $n$ at the four wavelengths of 405, 473, 532 and 633 nm ($n_{405}$, $n_{473}$, $n_{532}$ and $n_{633}$, respectively). In particular, we report comprehensive RI measurements for the hygroscopic response of aqueous droplets containing atmospherically relevant inorganic solutes and quantify the RI retrieval precision. We compare the differences in the RI retrieval precision and accuracy when sizing

particles using either (i) light elastically scattered from the BB optical trap, or (ii) light elastically scattered from a Gaussian probe beam. Measurements of $n_{633}$ as a function of RH from phase functions, Raman spectroscopy (from aerosol optical tweezer measurements) and from the Tang *et al.* (Tang et al., 1997; Tang and Munkelwitz, 1994) parameterisation are compared. Finally, our multi-wavelength RI retrievals, combined with $n_{650}$ measurements with aerosol optical tweezers, $n_{589}$ measurements on bulk solutions and the $n_{633}$ descriptions provided by Tang *et al*., (Tang et al., 1997; Tang and Munkelwitz,

1994) are parameterised using an optical dispersion model. The following section describes the 405-nm and 532-nm SP-CRDS instruments and the aerosol optical tweezers instrument, while Sect. 3 reports the retrieved RI variation with RH for aqueous aerosol particles containing the inorganic solutes NaCl, NaNO$_3$, (NH$_4$)$_2$SO$_4$, NH$_4$HSO$_4$ or Na$_2$SO$_4$ and compares different methods of retrieving $n_\lambda$. Indeed, these solutes represent the most abundant species found in inorganic atmospheric aerosol. Finally, Sect. 4 presents a Cauchy dispersion model for parameterising the variation in the RI with both wavelength

and RH.



## 2. Experimental and Numerical Methods

### Aerosol Optical Tweezers

Measurements of refractive index at 650 nm were accomplished using a commercial aerosol optical tweezers (AOT, AOT-

100, Biral). The experimental setup has been described in detail previously (Davies and Wilson, 2016; Haddrell et al., 2017). Briefly, a single aqueous droplet from a plume produced by a medical nebuliser (NE-U22, Omron) is captured by a gradient force optical trap formed from focussing a 532 nm laser (Opus 2W, Laser Quantum) through a high numerical aperture microscope objective (Olympus PLFLN 100×). Inelastically backscattered (Raman) light is imaged onto the entrance slit of a 0.5 m focal length spectrograph (Princeton Instruments, Action Spectra SP-2500i), dispersed by a 1200 line pairs per mm

grating onto a cooled CCD camera. The Raman spectrum of a spherical droplet consists of a broad underlying Stokes band with superimposed resonant structure at wavelengths commensurate with whispering gallery modes (WGMs), from which the radius, RI, and dispersion can be determined with accuracies better than 2 nm, 0.0005 and $3 \times 10^{-8}$ cm respectively (Preston and Reid, 2013). RH in the trapping chamber is controlled by adjusting the relative flows of dry and humidified nitrogen and is measured using a capacitance probe (Honeywell HIH-4602C) to ±2%. A typical experiment involves

trapping an aqueous droplet containing one of the studied solutes, decreasing the RH in discrete steps over several hours, and monitoring the RH-dependent changes to droplet radius, refractive index and dispersion. Note that because dispersion is also determined, it is possible to compare refractive indices measured using the optical tweezers with other approaches at different wavelengths (e.g. 633 nm by Tang *et al.*).

### Single Particle Cavity Ring-Down Spectroscopy Instrument

Figure 1 summarises the experimental arrangement of the 405-nm and 532-nm SP-CRDS instruments. The 532-nm SP-CRDS has been described previously (Mason et al., 2015), while Cotterell *et al.* (Cotterell et al., 2015a) describes modifications to the instrument to improve particle size retrieval. The 405-nm SP-CRDS is described elsewhere (Cotterell et al., 2015b), and recent modifications to this instrument to improve size determination are described below. The following

section provides a general description of $\sigma_{ext}$ and elastic light scattering measurements using SP-CRDS.

### 2.1.    Overview of Single Particle Cavity Ring-Down Spectroscopy

### The Continuous Wave Cavity Ring-Down Spectrometer

The single particle CRDS measurements were performed on two separate instruments summarised in Figure 1, with each instrument performing CRDS at either 405 or 532 nm. The beam from a continuous wave, single mode laser (with a spectral

bandwidth < 5 MHz) passes through an acousto-optic modulator (AOM). The first-order diffraction spot is injected into an optical cavity, while the zeroth and higher order spots are attenuated with beam blocks. The optical cavity consists of two highly reflective mirrors with radii of curvature of 1 m and reflectivities > 99.98 % at the CRDS wavelength, and has a free spectral range of ~ 300 MHz. The cavity is aligned such that the TEM$_{00}$ mode is preferentially excited. A piezo ring actuator,



affixed to the rear cavity mirror, continuously varies the cavity length across several free spectral ranges. When the cavity length is such that a longitudinal cavity mode is excited, a photodiode monitors the build-up of light inside the cavity and measures the intensity escaping from the rear mirror as an output voltage, $V$, which is sent to both a Compuscope digitizer and a digital delay generator. The digital delay generator outputs a 5 V TTL pulse to the AOM when the leading edge of the

photodiode signal reaches a 1 V threshold value, rapidly switching off the first-order AOM diffraction beam and initiating a ring-down decay. The subsequent time variation in $V$ obeys a single exponential decay and is fitted to $V = V_0 \exp(-t/\tau) + b$, with $\tau$ the characteristic ring-down time (RDT) and $b$ a baseline offset. RDTs are measured at a rate of 5 – 10 Hz. To reduce the contributions of airborne dust particles to light extinction and to prevent the mirrors getting dirty, nitrogen gas flows are directed across the faces of the cavity mirrors and through flow tubes that extend to a trapping cell at the centre of

the cavity.

### The Bessel Beam Optical Trap

Both the 405-nm and 532-nm SP-CRDS instruments use a 532 nm laser beam to generate a BB optical trap. A Gaussian 532 nm laser beam is passed through a 2° axicon to produce a BB, which has a circularly symmetric intensity profile consisting

of a central core and multiple rings. A pair of lenses reduces the BB core diameter to 3 - 5 µm. A 45° mirror propagates this beam vertically into the trapping cell. This mirror and the trapping cell are mounted on the same translation stage, allowing the position of the BB optical trap to be translated in the horizontal transverse direction to the CRDS optical axis. In the trapping cell, the radiation pressure exerted by the BB on a trapped particle is balanced by a humidified nitrogen gas flow of 100 - 200 sccm, allowing control over RH. This gas flow is also used to purge the cell of excess aerosol. The RH inside the

trapping cell is monitored using a capacitance probe. A plume of aerosol particles is introduced into the cell using a medical nebuliser. Typically, a single aerosol particle is isolated from this plume by the Bessel beam optical trap. Occasionally multiple particles are trapped, in which case the trapping cell is evacuated and aerosol particles are again nebulised into the cell. This process is repeated until only a single particle is trapped.

### Size and RI Retrieval using Phase Function Images

A camera coupled to a high numerical aperture objective, positioned at 90º with respect to the BB propagation direction, records images (referred to as *phase function*, PF, images) that describe the angular variation in elastically scattered light at a selected polarisation. For 532-nm SP-CRDS measurements, we record PFs of the light elastically scattered from a 473 nm probe laser beam. The probe beam is aligned collinearly to the BB propagation direction using a polarisation beam splitter

(PBS) cube, the alignment of which is shown in Figure 1(a). The PBS merges the two beams prior to the final lens ($f = 50$ mm) and, therefore, the probe beam is weakly focussed into the trapping cell. In the trapping cell, the probe beam has a Gaussian transverse intensity profile and a beam diameter $\sim 8\times$ larger than the BB core diameter. A 532-nm laser line filter ensures the camera does not collect elastically scattered light from the BB. For 405-nm SP-CRDS measurements, we





predominantly collect PFs of light elastically scattered from the beam forming the BB optical trap or occasionally from a 633 nm probe laser beam (LHP073, Melles Griot). The 633 nm probe beam is aligned collinearly to the BB propagation direction and is merged with the BB using a PBS cube as shown in Figure 1(b). In the trapping cell, the probe beam has a Gaussian transverse intensity profile and a beam diameter that is ~ 3× larger than the BB core diameter. When recording 633 nm PFs,

a 532-nm laser line filter ensures the imaging camera does not collect elastically scattered light from the BB, while this filter is removed when collecting 532 nm PFs corresponding to BB illumination.

The variation in particle radius and $n_\lambda$ are determined by fitting the complete measured PF data set to Mie theory in a self-consistent step. We previously reported the algorithms used to perform this fitting (Cotterell et al., 2015a, 2015b; Preston

and Reid, 2015). For all hygroscopic response measurements reported in this work, the RI varies with the particle size. Therefore, we parameterize $n_\lambda$ in terms of particle radius using the expression:

$$n_\lambda = n_{\lambda,0} + \frac{n_{\lambda,1}}{a^3} + \frac{n_{\lambda,2}}{a^6} \qquad\qquad 2$$

in which $a$ is the particle radius, $n_{\lambda,1}$ and $n_{\lambda,2}$ are fitting parameters, while $n_{\lambda,0}$ is the real RI of pure water at wavelength $\lambda$. This latter term is known precisely from bulk measurements and we use the data of Daimon and Masumura for water at T = 24 °C (Daimon and Masumura, 2007). Specifically, we use $n_{405,0} = 1.343$, $n_{473,0} = 1.338$, $n_{532,0} = 1.335$, $n_{633,0} = 1.332$ and

$n_{650,0} = 1.331$.

### *Measuring $\sigma_{ext}$ for a Single Particle*

Once a single aerosol particle is optically trapped, the particle position is optimised to obtain a minimum in the measured RDT, corresponding to the particle being located at the centre of the ring-down beam. The position is varied in both

transverse directions to the CRD beam, in the vertical direction by changing the laser power and in the horizontal direction by scrolling the translation stage on which the trapping cell is mounted. When the particle is centred in the CRD beam, a computer-controlled laser feedback is initiated to maintain a constant particle height over the duration of the measurement and values for $\tau$ are collected. After measurements have been made on the particle, the empty cavity RDT, $\tau_0$, is recorded over several minutes. From knowledge of $\tau$ and $\tau_0$, $\sigma_{ext}$ is calculated using:

$$\sigma_{ext} = \frac{L\pi w^2}{2c}\left(\frac{1}{\tau} - \frac{1}{\tau_0}\right) \qquad\qquad 3$$

in which $L$ is the separation distance between the two cavity mirrors, $c$ is the speed of light and $w$ is the focal beam waist of the intra-cavity ring-down beam, which is either calculated using Gaussian optics (Kogelnik and Li, 1966) or is treated as a variable parameter when fitting the $\sigma_{ext}$ data to the prediction of a light scattering model. We use the latter method for determining $w$ and have previously demonstrated that the fitted $w$ agrees with the predictions of Gaussian optics (Cotterell et al., 2015b; Mason et al., 2014, 2015)






### 2.2. Computational Analysis of $\sigma_{ext}$ Data

Mie theory assumes that a travelling plane wave illuminates a homogeneous spherical particle. However, the field inside the optical cavity is a standing wave and not a travelling wave, with the extinction of light by aerosol varying for different phases of the standing wave (Mason et al., 2014; Miller and Orr-Ewing, 2007). The two cases of a particle centred at a

standing wave node or anti-node provide limiting values for $\sigma_{ext}$. A confined particle undergoes Brownian motion within the BB core over micrometre distances, allowing the particle to sample standing wave nodes, anti-nodes and intermediary phases. Therefore, the standing wave leads to a broadening in the measured $\sigma_{ext}$, with the limits in the recorded data corresponding to the particle located at either a node or anti-node. To fit our measured $\sigma_{ext}$ vs radius data, we use the cavity standing wave generalised Lorenz-Mie theory (CSW-GLMT) equations that we have derived previously to calculate $\sigma_{ext}$ for

the limiting cases of a node and anti-node centred particle (Cotterell et al., 2016). These calculations provide a CSW-GLMT envelope, within which all the measured data are expected to lie (in the absence of experimental noise) for the best-fit simulation. A residual function, $R$, is defined by Eq. (4), in which the $\sigma_{ext,j}$ points included in the summation are only those with measured values that reside outside the simulated $\sigma_{sim}$ envelope and $\sigma_{sim,j}$ is the closer of the node or anti-node centered simulation values at the given radius.

$$R = \frac{1}{J} \sum_{j=1}^{J} \frac{\left| \sigma_{ext,j} - \sigma_{sim,j} \right|}{\rho_j} \qquad \qquad 4$$

The density of the measured $\sigma_{ext}$ data points within a 1 nm range of the particle radius, $\rho_j$, is used as a weighting factor and prevents biasing of the fit to regions where the measured number of data points is high in the radius domain. The RI is varied to fit CSW-GLMT to the $\sigma_{ext}$ data, with the RI parameterized in terms of particle radius by Eq. (2). The value of $R$ was minimized by varying the parameters $n_{\lambda,1}$ and $n_{\lambda,2}$, in addition to varying the beam waist, $w$. The $n_{\lambda,1}$ and $n_{\lambda,2}$ values that correspond to the minimum in $R$ define the best-fit refractive index. For the $1 - 2$ μm radius particles studied using either

405-nm or 532-nm SP-CRDS, $n_{\lambda,1}$ was typically varied over a range of 0 to $1 \times 10^8$ nm$^3$ in steps of $1 \times 10^6$ nm$^3$, while $n_{\lambda,2}$ was varied over a range of $-1 \times 10^{18}$ to 0 nm$^6$ in steps of $1 \times 10^{16}$. A grid search was used to vary the three parameters $n_{\lambda,1}$, $n_{\lambda,2}$, $w$ such that all points in the three dimensional search space were sampled.

### 3. SP-CRDS Measurements for Hygroscopic Inorganic Aerosol Particles

Single aqueous aerosol droplets containing one of the inorganic solutes of interest (NaCl, NaNO$_3$, (NH$_4$)$_2$SO$_4$, NH$_4$HSO$_4$ or Na$_2$SO$_4$) were optically trapped using the SP-CRDS instruments at high (~80 – 85 %) RH. Following nebulisation, the ambient RH was unsteady and so no action was taken during a conditioning period of ~ 10 minutes until the RH had stabilised. The RH was subsequently lowered over time at a near-constant rate of 0.4 – 0.5 % per minute, until the particle effloresced or the particle size was such that it was unstable and was ejected from the optical trap. While lowering the RH, $\tau$

and PFs were recorded at a rate of ~10 s$^{-1}$ and 1 s$^{-1}$, respectively. Repeat measurements (~5 droplets) were performed for all inorganic solutes of interest using both of the SP-CRDS instruments and, in the case of 405-nm SP-CRDS, using the 532 nm





BB for PF measurements. This work considers measurements of particles with mean particle radii > 1 μm only, as we have previously found that $n_\lambda$ retrievals for hygroscopic particles in this size range are accurate to < 0.001, while this accuracy degrades significantly for smaller particles (Cotterell et al., 2016).

### 3.1. Radius and Refractive Index Retrieval from Recorded Phase Functions

Prior to analysis of the $\sigma_{ext}$ data, the particle radius and $n_\lambda$ at the PF acquisition wavelength were determined from fitting the PFs to Mie theory (see Sect. 2.1). As described previously (Cotterell et al., 2015a, 2015b), a mean Pearson correlation coefficient $\bar{c}(n_\lambda)$ is used to quantify the level of agreement between the measured PF data set and the set of best-fit Mie theory simulations, with $\bar{c}(n_\lambda) = 1$ corresponding to perfect agreement and lower values of $\bar{c}(n_\lambda)$ indicative of a poorer fit. The entire PF data set is made up of typically 5000 PFs, each with a corresponding Pearson correlation coefficient value,

$c(n_\lambda)$, which each contribute to the overall mean Pearson correlation coefficient value, $\bar{c}(n_\lambda)$. Figure 2(a) shows the variation in the fitted Pearson correlation coefficient $c(n_{473})$, radius and $n_{473}$ with time (labelled as frame number, with PF frame acquisition rate of 1 s⁻¹) when fitting 473 nm PFs from a measurement on an aqueous NaNO₃ particle. Data are shown for the initial $n_{473}$ fitting parameter values ($n_{473,1} = 5.6 \times 10^7$ nm³, $n_{473,2} = 0.0$ nm⁶), for further optimised values ($n_{473,1} = 7.3 \times 10^7$ nm³, $n_{473,2} = 0.0$ nm⁶) and for the best-fit parameter values ($n_{473,1} = 1.221 \times 10^8$ nm³, $n_{473,2} = -7.585 \times 10^{15}$ nm⁶). Figure

2(b) shows how the mean correlation coefficient $\bar{c}(n_{473})$ varies with $n_{473,1}$ and $n_{473,2}$ on their initial search cycles. Further data points can be seen that correspond to refined grid-search cycles. There are clear maxima in $\bar{c}(n_{473})$ as $n_{473,1}$ and $n_{473,2}$ are varied, with the maximum having a value $\bar{c}(n_{473}) = 0.9951$.

For the SP-CRDS measurements reported in this work, each hygroscopic response measurement has a $\bar{c}(n_\lambda)$ associated with

the fitting of the PFs. When using 473 (or 633) nm probe laser beam illumination, values of $\bar{c}(n_{473})$ (and $\bar{c}(n_{633})$) averaged over all hygroscopic response measurements performed at the respective wavelengths were 0.993 ± 0.003 (and 0.99646 ± 0.00004) where the errors represent one standard deviation in the $\bar{c}(n_\lambda)$. We also collected PFs of the elastically scattered light from the BB illumination, which resulted in lower values of $\bar{c}(n_\lambda)$ compared to illumination from a Gaussian probe beam; the corresponding $\bar{c}(n_{532})$ from BB illumination was 0.985 ± 0.008. This lower $\bar{c}(n_\lambda)$ value indicates that the use of

PFs from illumination with the BB light (core of diameter ~ 5.5 μm) is detrimental to radius and $n_{532}$ determination with more accurate determinations arising when analysing the PFs from Gaussian illumination at 473 and 633 nm. The component wave vectors constituting the BB make large angles with respect to the optical propagation axis, while Mie theory assumes that a plane wave illuminates a spherical particle. The 473 and 633 nm probe beams have Gaussian profiles with spot diameters estimated to be ~ 30 and ~ 15 μm, respectively, at the particle trapping location and are more

representative of plane wave illumination compared to that provided by the trapping Bessel laser beams. Sect. 3.4 further analyses the consequential errors in the derived particle size and $n_{532}$ due to BB illumination.





### 3.2. Comparison of Refractive Index Retrieval Methods

The 633 nm probe beam was used for PF illumination to provide a comparison of the refractive index retrieval accuracy from PF imaging with AOT measurements (using Eq. 5) and measurements by Tang *et al*. (Tang et al., 1997; Tang and Munkelwitz, 1994) for the aqueous NaCl system only. For the final set of data presented in Sect. 4, the parameterisation by

Tang *et al*. is used to represent $n_{633}$ for all compounds. In addition, for the retrieval of RIs from the 405-nm SP-CRDS system, all droplet radii were retrieved from PFs using BB illumination at 532 nm rather than at 633 nm (Tang et al., 1997; Tang and Munkelwitz, 1994).

The refractive index from AOT measurements is retrieved from Raman spectroscopy measured at 650 nm. The fitting of the

Raman spectra yields not only $n_{650}$ but also dispersion terms, $m_1$ and $m_2$, which allow the refractive index to be calculated at alternative wavelengths:

$$n = n_0 + m_1(v - v_0) + m_2(v - v_0)^2 \qquad\qquad 5$$

Here, $n$ is the refractive index at the desired wavelength, $n_0$ is the refractive index at the measured wavelength, $n_{650}$ in this case, $v$ is the wavenumber of the desired wavelength and $v_0$ is the wavenumber at the measurement illumination wavelength.

Figure 3 shows that $n_{633}$ values retrieved from the AOT and PFs measurements are in good agreement. Furthermore, $n_{633}$ from both techniques are consistent with the parameterisation of $n_{633}$ from Tang *et al*., within the RH uncertainty indicated corresponding to ± 2 % for AOT and PF measurements. Hargreaves *et al*. compare EDB and optical tweezers measurements of aqueous NaCl to the Tang parameterisation, reporting good agreement to Tang *et al.* at high RH (> 70 ± 0.2 % ) with increasing divergence at low RH (45 + 5 %), where the error in RH represents the equivalent offset in RH required to bring

the different measurements into agreement (Hargreaves et al., 2010). For $(NH_4)_2SO_4$, Tang *et al*. (Tang and Munkelwitz, 1994) acknowledged that although their measurements agreed with those of Richardson and Spann (Richardson and Spann, 1984) at high RH, there was divergence from measurements by Cohen and co-workers by an RH equivalent of 4 – 5 % (Cohen et al., 1987). Despite this, the $n_{633}$ measurements from three different techniques show good agreement within the uncertainty in RH, reinforcing the premise that the different techniques offer an accurate means of retrieving $n_\lambda$ as a function

of RH and are compatible for combining in the refractive index parameterisation presented in Sect. 4. This comparison also demonstrates that PF measurements can be used to accurately retrieve particle radius and refractive index. It should also be noted that the AOT measurements are centred at a wavelength of 650 nm and the values reported in the figure are the corrected values for 633 nm. Over this wavelength range, the RI change from 650 to 633 nm is ~ - 0.0017 at a typical RH of 80 %.






### 3.3.    Extinction Cross Section Measurements

Once the time-evolution in particle radius was determined from the PFs, the measured $\sigma_{ext}$ data were compared to CSW-GLMT calculations using the procedure described in Sect. 2.2. Figure 4 shows example $\sigma_{ext}$ data sets measured using either 405-nm or 532-nm SP-CRDS and the best-fit CSW Mie envelopes. For the example data presented in this figure, the particle

radii for measurements using 405-nm SP-CRDS are from fitting the 532-nm PFs (i.e. using BB illumination) and for the 532-nm SP-CRDS the radii are from fitting 473-nm PFs (from the probe beam illumination). The $\sigma_{ext}$ data sets measured using 405-nm SP-CRDS have a higher number of resonance features compared to the data sets measured using 532-nm SP-CRDS, even though the particles evaporated over similar radius ranges, owing to the $1/\lambda$ dependence of the particle size parameter. The higher number of resonance features in the $\sigma_{ext}$ data at lower wavelengths is expected to facilitate higher

precision determinations of $n_\lambda$.

### 3.4.    Multi-Wavelength Determinations of the Refractive Index

The precision in $n_\lambda$ retrievals from $\sigma_{ext}$ and PF data was tested by repeat SP-CRDS measurements on different aqueous droplets containing the inorganic solutes of interest. The retrieved $n_\lambda$ were represented as a function of the RH, as measured by a calibrated capacitance probe located $\sim 1$ cm from the droplet trapping location. The $n_\lambda$ data were binned in 2 % RH

intervals because the capacitance probe RH measurements have a standard error of $\pm 2$ %. Repeat measurements of $n_\lambda$ were then averaged for each RH bin and a standard deviation, $s_\lambda(RH)$, was calculated. The values of $n_{405}$ and $n_{473}$ were retrieved from fitting the $\sigma_{ext}(\lambda = 405$ nm) and PF($\lambda = 473$ nm) data, respectively. This section presents the average $n_{532}$ calculated from $\sigma_{ext}(\lambda = 532$ nm) data only and neglects the $n_{532}$ retrieved from PF analysis, assuming that the RI retrieval using $\sigma_{ext}$ data is more precise than using PFs from BB scattering. Sect. 3.5 presents a thorough analysis of the impact of Bessel beam

illumination on $n_{532}$ precision. Table 1 reports the number of particles studied for each inorganic species using either 405-nm or 532-nm SP-CRDS. Data are shown for 405-nm SP-CRDS measurements using the BB probe beam for PF collection.

Figure 5 summarises the average retrieved RIs ($n_{405}$, $n_{473}$ and $n_{532}$) binned in 2 % RH intervals for each of the inorganic solutes of interest. Also shown are the $n_{633}$ variations reported by Tang *et al*. (Tang et al., 1997; Tang and Munkelwitz, 1994)

and values for $n_{589}$ reported in the CRC Handbook of Chemistry and Physics (Haynes, 2015). The CRC handbook reports measured values of $n_{589}$ in terms of mass fraction of solute. Therefore, the E-AIM model was used to relate solute mass fraction to water activity (Clegg et al., 1998; Clegg and Wexler, 2016). Furthermore, the CRC Handbook does not report RI values for $NH_4HSO_4$ and the $n_{589}$ values plotted in Figure 5(d) and Figure 8(d) were taken from bulk solution measurements using a refractometer (Misco, Palm Abbe).


In general, Figure 5 shows that $n_\lambda$ increases towards shorter wavelengths, as expected given the chromatic dispersion behaviour of typical materials. Furthermore, the optical dispersion increases as the RH decreases and the solute becomes more concentrated within the aqueous droplets. The increasing separation between the measured RH-dependent $n_\lambda$ values for





progressively shorter wavelengths indicates the increasing influence of optical dispersion. At higher RH values (> 80 %), the measured values of $n_{405}$, $n_{473}$ and $n_{532}$ approach one-another and, in the case of NaNO$_3$, the values of $n_{405}$ become lower than the measured $n_{473}$ and $n_{532}$ values. The RI values are expected to converge as the RH tends to 100% (the RI variation with wavelength in the visible spectrum is low for pure water) (Daimon and Masumura, 2007), but this crossing is likely to derive

from calibration errors in the RH probes, noting that the set of $\{n_{405}, n_{532}\}$ and $\{n_{473}, n_{532}\}$ measurements are made using different SP-CRDS instruments.

The $n_{589}$ bulk measurements are limited by the solubility of each solute, which constrains measurements to high water activity (RH) values. As expected, in all cases, the bulk $n_{589}$ literature data lie close to, or match, the $n_{633}$ values. Small

deviations of $n_{589}$ from $n_{633}$ (> 0.003) for NH$_4$HSO$_4$, and NaNO$_3$ can be observed in Figure 5. While the RH values for the $n_{589}$ data are calculated from measured solute mass fraction data (which are expected to be accurate to < 1%) using the E-AIM model, (Clegg et al., 1998; Clegg and Wexler, 2016) the RH measurements associated with the $n_{633}$ parameterisation are expected to be more uncertain, to ± 5 % at RH < 45 %, as discussed in Sect. 3.2. Therefore, it is reasonable to see deviations in the order of the data points at high RH in line with the uncertainty associated with the RH measurements.

In most cases the $n_{650}$ data lie below the data measured at shorter wavelengths, which follows the expected trend of chromatic dispersion. In the cases of NaNO$_3$, (NH$_4$)$_2$SO$_4$ and NH$_4$HSO$_4$ the $n_{650}$ values cross the $n_{633}$ values, which is attributed to uncertainties in RH measurements from both the AOT and Tang parameterisation of ± 2 % and ∼ ± 5 % respectively (Tang and Munkelwitz, 1994).

**3.5.    Precision in SP-CRDS Determined Refractive Index**

The precision in the SP-CRDS determined RI values can be quantified by the standard deviation in $n_\lambda$ within a 2% RH interval. For ease of reading, the plots in Figure 5 do not show error bars representing this standard deviation. Instead, the means of the standard deviation values over all the RH bins are calculated and denoted $\overline{s(n_\lambda)}$. Table 1 presents the $\overline{s(n_\lambda)}$ values for each inorganic solute of interest as measured on each of the SP-CRDS instruments. In all cases, the retrieved RI

from fitting $\sigma_{ext}$ data is more precise than from the corresponding PF analysis.

For measurements performed using the 532-nm SP-CRDS instrument, the precisions of $n_{473}$ and $n_{532}$ measurements (from PFs and CRDS respectively) were generally high, indicated by the majority of $\overline{s(n_{473})} < 0.004$ and $\overline{s(n_{532})} < 0.003$ in Table 1. The only exception was Na$_2$SO$_4$ where the standard deviations for the measurements are larger, $\overline{s(n_{473})} \sim 0.007$

and $\overline{s(n_{532})} \sim 0.006$. Na$_2$SO$_4$ effloresces at relatively high RH (∼ 60 %), thereby limiting the radius range accessed during drying. This reduces the extent of the resonant structure observed, which provides the greatest constraint on RI retrievals, and limits the precision of the $n_\lambda$ determination. The measurements from the 405-nm SP-CRDS are not as precise as those performed on the 532-nm SP-CRDS, owing to the collection of PFs from BB illumination, where $\overline{s(n_{532})}$ and $\overline{s(n_{473})}$



values are in the range of 0.005 – 0.013 and 0.003 – 0.007, respectively. The $\overline{s(n_{532})}$ values from (BB illuminated) PFs are significantly higher than $\overline{s(n_{405})}$ retrieved from CRDS; for the majority of solutes (except NH$_4$HSO$_4$) having $\overline{s(n_{405})} <$ 0.0045. The NH$_4$HSO$_4$ $\overline{s(n_{532})}$ values are particularly high, which influences the retrieved $\overline{s(n_{405})}$, since information relating to the geometric size of the particle from PF data is required in the $\sigma_{ext}$ data fitting procedure. The NH$_4$HSO$_4$ values

of $\overline{s(n_{532})} = 0.013$ and $\overline{s(n_{405})} \sim 0.01$ emphasise the recommendation to use a probe Gaussian beam for PF illumination to improve precision in both $n_\lambda$ from PFs and $n_\lambda$ from CRDS in the long term.

Figure 6 compares $n_{532}$ retrieved from CRDS and BB-illuminated PFs, further highlighting the improvements that can be achieved by adding a PF-probe beam. 405-nm SP-CRDS measurements were performed for aqueous Na$_2$SO$_4$ but the non-

linear dependence of $n_\lambda$ on RH and possible impurities in the sample prevented fitting the 532-nm PFs and therefore 405-nm $\sigma_{ext}$ measurements to Mie Theory and CSW-GLMT, respectively. Therefore, Na$_2$SO$_4$ $n_{405}$ measurements are not discussed further. For each inorganic solute studied, the precision in the $n_{532}$ values retrieved from the PFs is poorer than retrievals from $\sigma_{ext}$ data, indicated by PF $\overline{s(n_{532})}$ values being approximately twice the equivalent $\overline{s(n_{532})}$ $\sigma_{ext}$ data, as indicated in Table 1. Moreover, the slopes of the RI vs RH plots in Figure 6(a) – (d) are steeper for the PF-retrieved, compared to the $\sigma_{ext}$-

retrieved, $n_{532}$ data. The PF-retrieved $n_{532}$ for NH$_4$HSO$_4$ (Figure 6(d)) has both significantly larger standard deviation and divergence from $n_{532}$ $\sigma_{ext}$ data. This uncertainty is evident in the $\sigma_{ext}$-retrieved $n_{405}$ values, with $\overline{s(n_{405})} = 0.0095$. Although the precisions of $n_{532}$ and radius retrievals are compromised when fitting PFs using BB illumination, the corresponding $\overline{s(n_{405})}$ from fitting $\sigma_{ext}$ data is, on the whole, close to $\sigma_{ext}$-retrieved $\overline{s(n_{532})}$ from probe beam PF illumination (with the exception of NH$_4$HSO$_4$), with values in the range of 0.003 – 0.005 (Table 1). Furthermore, the $\overline{s(n_{405})}$ values indicate a

precision better than twice that of the corresponding $\overline{s(n_{532})}$ from fitting PFs.

All $\overline{s(n_\lambda)}$ values encompass contributions from both variation in $n_\lambda$ and, more significantly, from uncertainties in measuring the RH. Indeed, we have recently reported that the $n_\lambda$ retrieval accuracy from fitting 532-nm SP-CRDS $\sigma_{ext}$ data is $< \pm 0.001$ for the size range of hygroscopic response measurements performed in this paper ($> 1$ µm), although our previous analysis

neglected the influence of sample impurities on the retrieved $n_\lambda$ (Cotterell et al., 2016). As an example, $\overline{s(n_{473})}$ and $\overline{s(n_{532})}$ take respective values of 0.0028 and 0.0015 for the measurements on NaNO$_3$. The uncertainty of $\pm 2$ % in the recorded RH probe value is calculated to contribute uncertainties of 0.0023 and 0.0021 to $\overline{s(n_{473})}$ and $\overline{s(n_{532})}$, respectively. Therefore, any errors in particle sizing from 473-nm PFs and noise in the recorded $\tau$ data make only a small contribution to the $n_\lambda$ precision in the RH domain.






### 4. Parameterising the Refractive Index with Variation in Wavelength and RH

The plots in Figure 5 contain all the information to characterise the RI in terms of both wavelength and RH. Here, we develop a refractive index model that accounts for variations in both wavelength and RH, and this model is fitted to the data in Figure 5 to parameterize $n(\lambda, RH)$. The Cauchy equation is an empirical relation describing the wavelength dependence in

the refractive index, and can be written as (David et al., 2016):

$$n = n_0 + \sum_{i=1}^{N} n_i \left[ \left( \frac{\lambda_0}{\lambda} \right)^{2i} - 1 \right] \qquad\qquad 6$$

in which $n_0$ is the refractive index at reference wavelength $\lambda_0$ and $n_i$ are dispersion coefficients. In our model, we find that expansion of the summation in Eq. (6) to $i = 1$ is required only:

$$n = n_0 + n_1 \left[ \left( \frac{\lambda_0}{\lambda} \right)^{2} - 1 \right] \qquad\qquad 7$$

To incorporate the RH dependence of the refractive index into Eq. (7), we note that $n_0$ and $n_1$ are expected to be smooth functions of RH and so we parameterize $n_0$ and $n_1$ as polynomial functions of the water activity. In our Cauchy model, we

use $\lambda_0 = 525$ nm, since this wavelength is at the centre of the wavelength range (405 – 650 nm) over which we have RI data available. The values for $n_0$ and $n_1$ are described by the following quartic and cubic polynomial equations respectively in terms of $a_w$ :

$$n_0 = n_{0,0} + n_{0,1}(100a_w) + n_{0,2}(100a_w)^2 + n_{0,3}(100a_w)^3 + n_{0,4}(100a_w)^4 \qquad\qquad 8$$

$$n_1 = n_{1,0} + n_{1,1}(100a_w) + n_{1,2}(100a_w)^2 + n_{1,3}(100a_w)^3 \qquad\qquad 9$$

The parameters $(n_{0,x},\ldots)$ and $(n_{1,x},\ldots)$ are fitted concurrently by performing a least squares fit to Eq. (7) by minimising the residual between the measured and literature data ($n_{405}$, $n_{473}$, $n_{532}$, $n_{650}$, $n_{589}$, $n_{633}$) and the Cauchy model generated $n_\lambda$. The

Microsoft Excel GRG non-linear engine is used to accomplish this fitting, constraining the modelled $n_\lambda$ to the values of pure water at RH = 100% (Daimon and Masumura, 2007). Figure 7 presents the results as contour plots for each inorganic solute of interest. These contour plots represent the most comprehensive description of refractive index for atmospherically relevant inorganic aerosol, fully characterising the RI variation with both visible wavelength and RH. The best-fit parameters for Eq. (8) and (9) that describe these contour plots are summarised in Table 2.

Figure 8 compares the measured data points with the Cauchy model curves from the aforementioned global fitting procedure (i.e. using the relevant parameters in Table 2). This plot only shows data at 10% RH intervals for simplicity. The total number of literature and measured data points, $N$, used in the fitting procedure are indicated for each solute in Table 2. The agreement between the Cauchy model and data at RH = 100% highlights that the model is constrained to the RI of pure

water. The error bars represent the ± 2 % uncertainty associated with RH measurements in the $n_\lambda$ domain. Within these error bars, there is generally good agreement between the measured data and the Cauchy model. In all cases (except NH$_4$HSO$_4$, as





previously discussed) the Cauchy model describes the measured $n_{405}$, $n_{473}$ and $n_{532}$ data well. There is good agreement between $n_{633}$ data and the Cauchy model at high RH. However, the literature $n_{633}$ values are lower than the calculated $n_{633}$ for NaCl and $(NH_2)_4SO_4$ at low RH. These discrepancies are attributed to the reduced number of literature and measured data points for $\lambda > 600$ nm at the low RHs. The bulk $n_{589}$ data points are described well by the Cauchy model; two of the NH$_4$HSO$_4$ measured $n_{589}$ values are lower than the modelled $n_{589}$. The $n_{650}$ values from AOT measurements lie marginally lower than is expected from the Cauchy model. This is also evident in Figure 3 where AOT measurements (calculated at $n_{633}$) were compared to those parameterised by Tang *et al.*, and implies the RHs reported for each of the $n_{650}$ AOT measurements are systematically low by a value close to the - 2 % uncertainty associated with the RH measurements.

The overall agreement between the global Cauchy fit ($n_{fit}$) and the measured and literature values ($n_j$) is quantified by evaluating the mean refractive index difference:

$$\overline{|\Delta n|} = \frac{1}{J} \sum_{j=1}^{J} |n_{fit} - n_j| \qquad\qquad 10$$

in which the summation is over all $J$ measured/literature values. Table 2 summarises the values of $\overline{|\Delta n|}$ for the inorganic solutes studied here. The solutes NaCl, NaNO$_3$ and $(NH_4)_2SO_4$ have $\overline{|\Delta n|} \leq 0.002$ which indicates the measured and literature data points lie very closely to the Cauchy parameterization. The solutes Na$_2$SO$_4$ and NH$_4$HSO$_4$ give $\overline{|\Delta n|} \approx 0.003$ and $\overline{|\Delta n|} \approx 0.0044$ respectively. The limited number of measured and literature data for Na$_2$SO$_4$ is a consequence of the high efflorescence RH and results in a less well constrained refractive index description, which is reflected in the $\overline{|\Delta n|}$ value.

## 5. Conclusions

Comprehensive $\sigma_{ext}$ and PF measurements are reported for the hygroscopic response of aqueous droplets containing inorganic solutes of NaCl, NaNO$_3$, $(NH_4)_2SO_4$, NH$_4$HSO$_4$ or Na$_2$SO$_4$, using either 405-nm or 532-nm SP-CRDS. These measurements permitted RI retrieval at wavelengths of 405, 473 and 532 nm. The $n_\lambda$ retrieved from PFs when using a Gaussian laser beam, at 473 nm or 633 nm with a ~ 30 μm or ~15 μm beam waist, respectively, had twice the precision compared to PFs illuminated with a BB (at 532 nm). The addition of a probe Gaussian-profile laser beam for retrieving particle size and $n_\lambda$ from PFs improves the precision and accuracy of $n_\lambda$ retrievals from $\sigma_{ext}$ measurements. Importantly, the precision in all RI retrievals is considerably better that the typical ± 0.02 precision of aerosol ensemble CRDS measurements (Mason et al., 2012).

A Cauchy dispersion model provided an effective parameterisation of our measured (and literature) refractive index data in terms of both wavelength, from 400 – 650 nm, and RH, 100 % to efflorescence RH (Eq. (7) – (9)), with Table 2 reporting the best-fit coefficients. This represents the most precise and comprehensive description of refractive index of inorganic atmospheric aerosol to date. These parameterisations will be useful to researchers wishing to calculate the refractive indices





of aerosol containing the main inorganic species found in atmospheric aerosol, for example in the calculation of radiative forcing efficiency or in the interpretation of measurements made by optical instruments.

**Author contribution**

M. I. Cotterell and R. E. Willoughby contributed equally to the data presented in this work. J. P. Reid and A. J. Orr-Ewing designed the instruments described in this work. B. R. Bzdek assisted with optical tweezer measurements and all co-authors contributed to the preparation of the manuscript.

**Acknowledgements**

JPR acknowledges financial support from the EPSRC through a Leadership Fellowship (EP/G007713/1). MIC acknowledges funding from NERC (NE/J01754X/1) and the RSC through an Analytical Trust Fund studentship and support from the Aerosol Society in the form of a CN Davies award. REW acknowledges funding from NERC GW4+ DTP (NE/L002434/1) and support from the Aerosol Society in the form of the CN Davies award. BRB acknowledges support from the ESPRC
(EP/L010569/1).

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





**TABLES**

|  | 405-nm SP-CRDS (using 532-nm PFs) | | | 532-nm SP-CRDS | | |
|---|---|---|---|---|---|---|
|  | $N$ | $\overline{s(n_{532})}$ (PFs) | $\overline{s(n_{405})}$ ($\sigma_{ext}$) | $N$ | $\overline{s(n_{473})}$ (PFs) | $\overline{s(n_{532})}$ ($\sigma_{ext}$) |
| NaCl | 5 | 0.0088 | 0.0044 | 7 | 0.0038 | 0.0030 |
| NaNO$_3$ | 7 | 0.0053 | 0.0034 | 5 | 0.0028 | 0.0015 |
| (NH$_4$)$_2$SO$_4$ | 4 | 0.0086 | 0.0041 | 9 | 0.0036 | 0.0027 |
| NH$_4$HSO$_4$ | 3 | 0.0130 | 0.0095 | 4 | 0.0054 | 0.0025 |
| Na$_2$SO$_4$ | - | - | - | 9 | 0.0072 | 0.0062 |

**Table 1:** Information relating to the precision of $n_\lambda$ measured from SP-CRDS for aqueous NaCl, NaNO$_3$, (NH$_4$)$_2$SO$_4$, NH$_4$HSO$_4$ and Na$_2$SO$_4$ droplets. $N$ is the number of data sets available for phase function and extinction cross section ($\sigma_{ext}$) analysis from either 405-nm or 532-nm SP-CRDS. The mean standard deviation in the retrieved $n_\lambda$, $\overline{s(n_\lambda)}$, is reported for both extinction cross section and phase function (PF) measurements.

|  | NaCl | NaNO$_3$ | (NH$_4$)$_2$SO$_4$ | NH$_4$HSO$_4$ | Na$_2$SO$_4$ |
|---|---|---|---|---|---|
| $n_{0,0}$ | 1.3495 | 1.4767 | 1.3764 | 1.4649 | 2.3810 |
| $n_{0,1}$ / $10^{-2}$ | 0.8770 | -0.0761 | 0.8552 | 0.0690 | -3.2494 |
| $n_{0,2}$ / $10^{-4}$ | -2.6190 | -0.1451 | -2.7632 | -0.4877 | 3.3485 |
| $n_{0,3}$ / $10^{-6}$ | 2.8864 | 0.2941 | 3.3769 | 0.6234 | -0.5428 |
| $n_{0,4}$ / $10^{-8}$ | -1.1586 | -0.2147 | -1.5096 | -0.3336 | -0.6018 |
| $n_{1,0}$ / $10^{-2}$ | 7.0981 | 3.1219 | -5.7022 | 3.8202 | 34.8882 |
| $n_{1,1}$ / $10^{-4}$ | -21.7961 | -1.8033 | 46.6467 | 4.1621 | -61.1350 |
| $n_{1,2}$ / $10^{-6}$ | 31.1422 | -1.4789 | -82.4706 | -6.4225 | 27.6052 |
| $n_{1,3}$ / $10^{-8}$ | -15.5951 | 1.3676 | 42.4736 | -0.5106 | 0.0000 |
| $N$ | 121 | 190 | 130 | 164 | 86 |
| $\overline{\lvert\Delta n\rvert}$ | 0.0020 | 0.0016 | 0.0018 | 0.0044 | 0.0027 |

**Table 2:** Summary of the best-fit parameters to describe the RI variation with both wavelength and RH for the inorganic solutes of interest. The parameters and function forms are defined in Eq. (7) – (9). $N$ is the total number of measured and literature data points (shown in Figure 5) to which the parameterisation was fitted. $\overline{\lvert\Delta n\rvert}$ is the mean absolute difference between the best-fit Cauchy model and the measured and literature data.




**FIGURES**

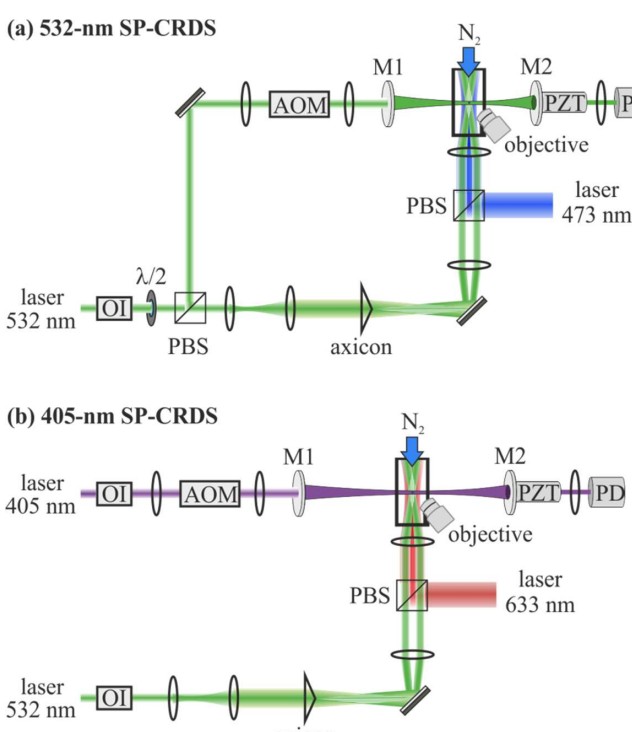

**Figure 1:** Schematic diagrams of the two SP-CRDS instruments used in this work. OI is an optical isolator, $\lambda/2$ is a half-wave plate, PBS is a polarising beam splitter cube, AOM is an acousto-optic modulator, PZT represents a piezo ring actuator, PD is a photodiode, M1 and M2 are highly reflective mirrors of the optical cavity. The 633 nm laser in Figure 1 (b) is used in selected measurements indicated in the text.



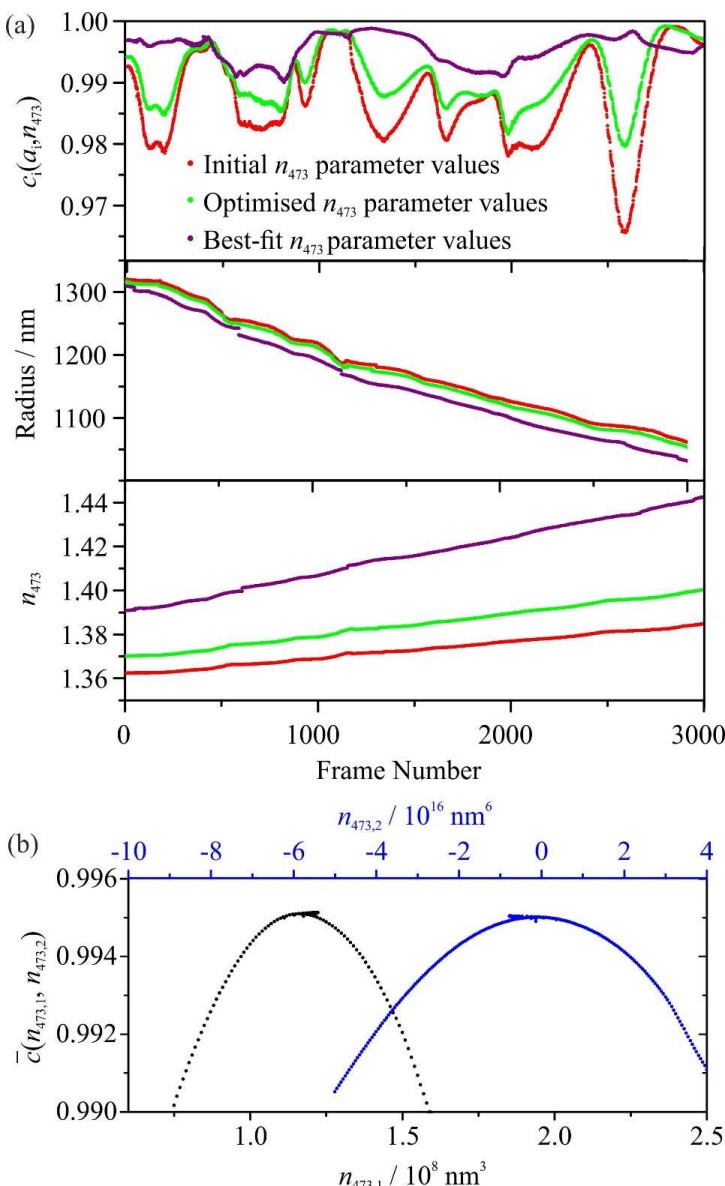

**Figure 2:** (a) The fitted radius, $n_{473}$, and correlation coefficient variation with frame number for an aqueous $NaNO_3$ droplet. Red points correspond to initial $n_{473}$ parameter values, green points to further optimised values and purple points to the best-fit values. (b) The mean correlation coefficient, $\bar{c}(n_{473})$, as the parameters $n_{473,1}$ and $n_{473,2}$ were varied.





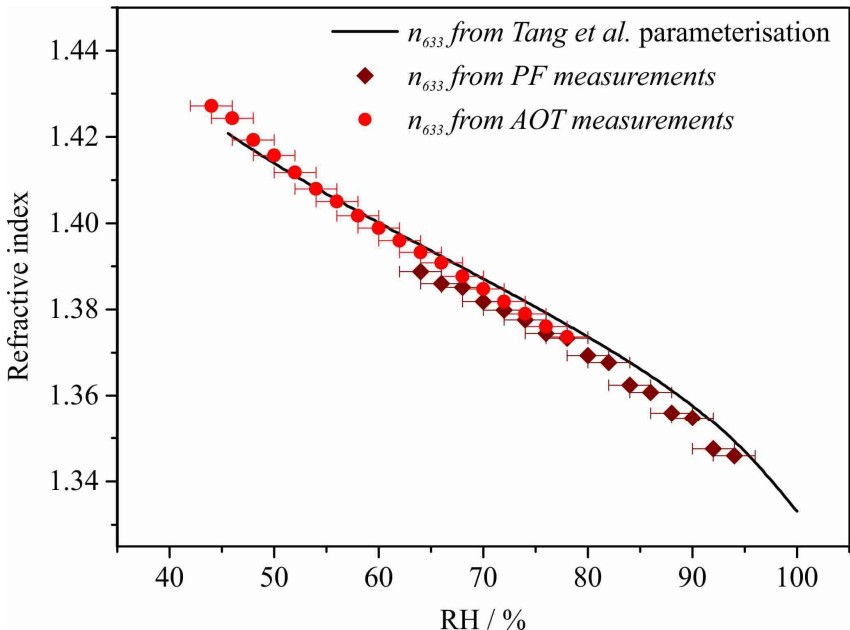

**Figure 3:** Comparison of the RH-dependant parameterisation of $n_{633}$ reported by Tang *et al.* (Tang et al., 1997; Tang and Munkelwitz, 1994) for aqueous NaCl to $n_{633}$ measured from 633 nm Gaussian-illuminated PFs and Raman spectroscopy from aerosol optical tweezers measurements. The error bars correspond to uncertainties associated with RH measurements.





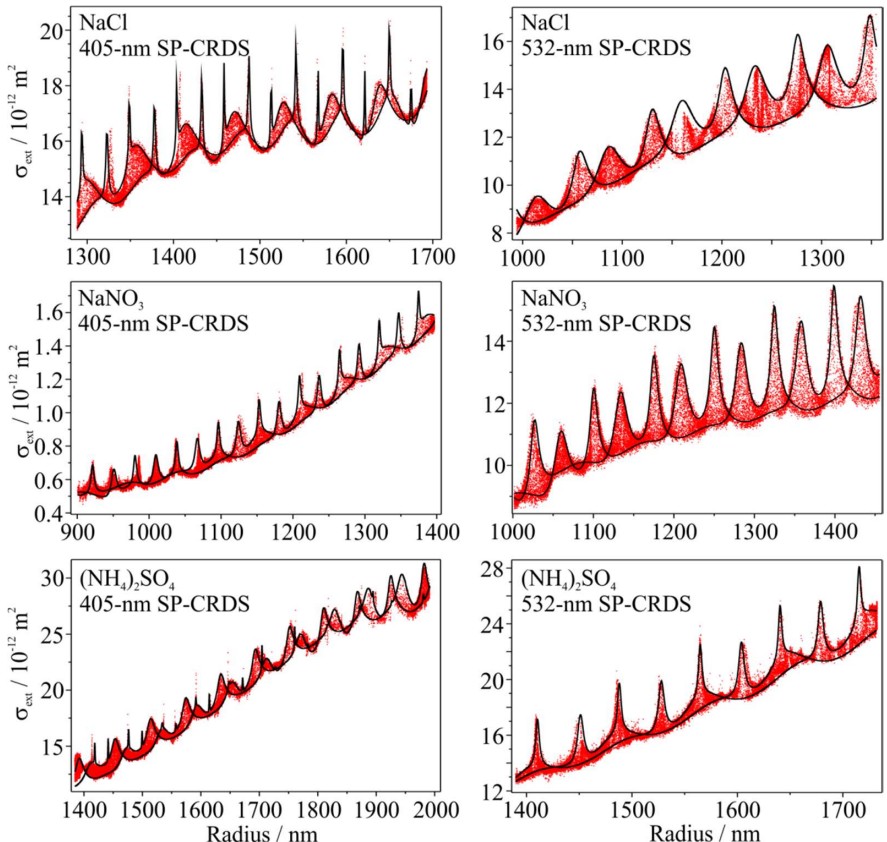

**Figure 4:** Representative SP-CRDS measurements of $\sigma_{ext}$ (red points) and best-fit CSW Mie envelopes (black lines) for aqueous aerosol particles containing the inorganic solutes NaCl, NaNO₃, and (NH₄)₂SO₄. The measured $\sigma_{ext}$ were collected using either 405-nm (left column) or 532-nm (right column) SP-CRDS.





**Figure 5:** The average $n_\lambda$ variations with RH for aqueous droplets containing the inorganic solutes of interest. The uncertainties in $n_\lambda$ have not been included here for clarity; they are listed in Table 1.





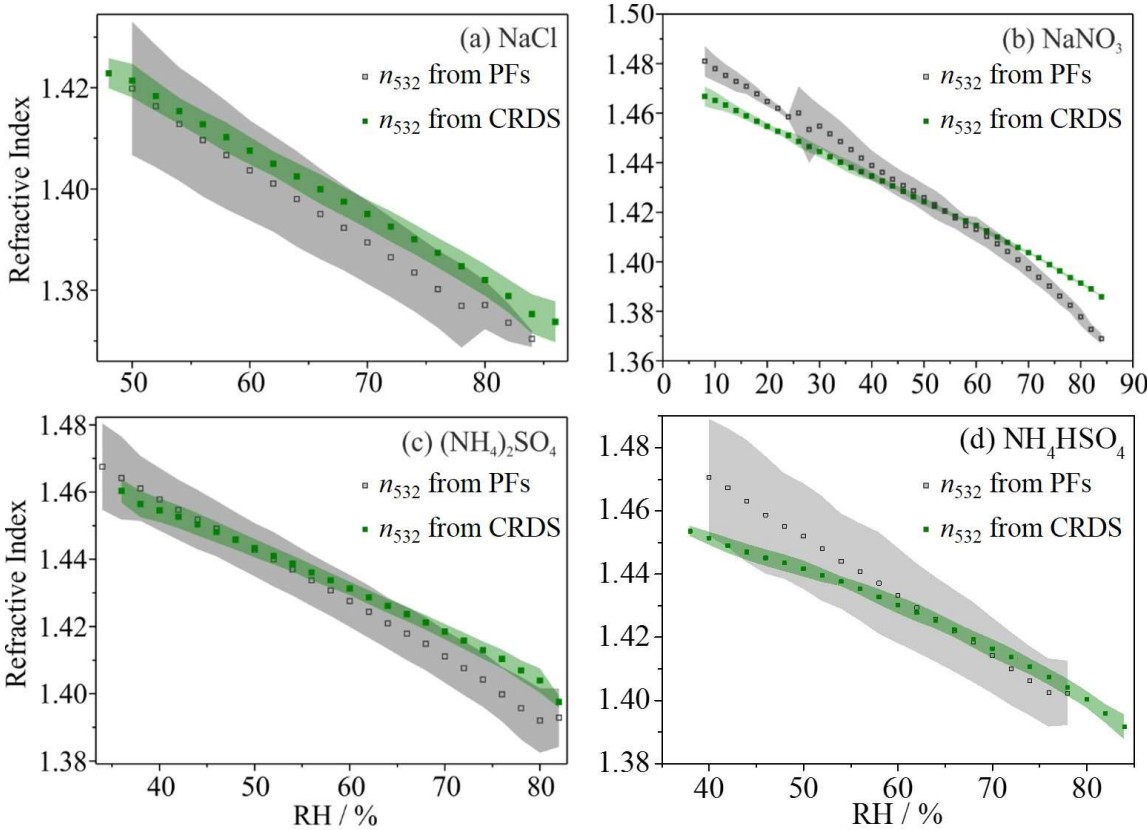

**Figure 6:** Comparison of $n_{532}$ values determined from either PFs recorded using BB illumination or $\sigma_{ext}$ data using 532-nm
5    SP-CRDS. The shaded envelopes represent one standard deviation in the measurements.

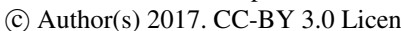



**Figure 7:** Contour plots representing the parameterisation of RI as a function of both wavelength and RH for aqueous aerosol particles containing the inorganic solutes of interest.







**Figure 8:** The best-fit Cauchy dispersion curves (solid lines) to measured and literature RI data (points) shown at 10 % RH
5    intervals (labels). The parameterisations were fitted to data at 2 % RH intervals, and the total numbers of data points
included in the parameterisation are specified in Table 2. The curves are found by a global fit of the Cauchy dispersion
model (Eq. (7) – (9)) to the measured and literature data in Figure 5 using the procedure described in the text. The error bars
represent the uncertainty in the RH measurements, indicated in the text, in the RI domain.