# Peer review of "A Complete Parameterization of the Relative Humidity and Wavelength Dependence of the Refractive Index of Hygroscopic Inorganic Aerosol Particles"

_Atmospheric Chemistry and Physics, 2017_

## Referee Comment (RC1) · Anonymous Referee #1 · 3 Apr 2017

Review of "A Complete Parameterization of the Relative Humidity and Wavelength Dependence of the Refractive Index of Hygroscopic Inorganic Aerosol Particles" by Cotterell et al.

This study reports on measurements using single particle cavity ring-down spectroscopy to derive refractive index values as a function of both wavelength (400 – 650 nm) and relative humidity for numerous inorganic solutes of atmospheric relevance. The topic of this paper is of relevance to this journal and it is of important to the research community as information related to refractive index is critical for quantifying

aerosol radiative forcing and various optical properties.

The paper is written very well and easy to follow. The title is reflective of the manuscript's content. The methods applied were solid and the conclusions reached are supported by the data. Tables and Figures are presented nicely. I find this piece of work to be a nice contribution to the literature relevant to refractive index and recommend its publication.
* * *

---

## Referee Comment (RC2) · Anonymous Referee #2 · 8 May 2017

I have read the article "A Complete Parameterization of the Relative Humidity and Wavelength Dependence of the Refractive Index of Hygroscopic Inorganic Aerosol Particles" by the authors Cotterell et al. submitted to Atmos. Chem. Phys. Discuss. This paper provides the refractive indices for aerosol particles composed of salt and water using several atmospherically relevant salts as a function of relative humidity from 100% RH to the efflorescence of the salt. Several wavelengths in the visible were used and dispersion coefficients were derived to obtain the refractive index at intermediate values. The significance of the article is summarized in section 4 "these ...[data] represent the most comprehensive description of refractive index for atmospherically

relevant inorganic aerosol, fully characterizing the RI variation with both visible wavelength[s] and RH". I have only a few minor comments that should be addressed.

The introduction is very complete and gives good motivation for the current study.

How does phase function from this system relate to phase function measured from a polar nephelometer? Could raw data be shown?

Why are the authors using one apparatus for one RH range and another for the higher RH range?

I struggled at times to read this article due to the many, many acronyms. For example, in other papers on optical properties, "BB" means broadband, and here it means Bessel beam, which leads to confusion. Please cut down on their use.

Fig 5: It is unclear why results vary so much from one another and are different than Tang et al. even for measurement at close to the same wavelength. I would expect less dispersion at larger wavelengths, but the differences look comparable in the 600nm region to the data for 405 vs 473 nm even though they are only separated by 13 nm.

Fig 6: Why is a large deviation seen for ammonium bisulfate at low RH between values obtained with CRD and those obtained from the phase function?

How easily expandable are the techniques used in this paper to the study of mixed organic/inorganic mixtures?

---

## Author Comment (AC1) · 4 Jul 2017

*Reviewer #1*
*This study reports on measurements using single particle cavity ring-down spectroscopy to derive refractive index values as a function of both wavelength (400 – 650 nm) and relative humidity for numerous inorganic solutes of atmospheric relevance. The topic of this paper is of relevance to this journal and it is of important to the research community as information related to refractive index is critical for quantifying aerosol radiative forcing and various optical properties.*

*The paper is written very well and easy to follow. The title is reflective of the manuscript's content. The methods applied were solid and the conclusions reached are supported by the data. Tables and Figures are presented nicely. I find this piece of work to be a nice contribution to the literature relevant to refractive index and recommend its publication.*

**Response**
We thank the reviewer for their positive feedback concerning our methods, manuscript preparation and the impact of our results.

---

## Author Comment (AC2) · 4 Jul 2017

**Response to Anonymous Referee #2**

The authors would like to thank Anonymous Reviewer #2 for their generally positive comments on the manuscript. We respond to the specific comments made by the referee below and identify the changes we have made to the manuscript.

**Specific comments:**

*Reviewer #2*
*I have read the article "A Complete Parameterization of the Relative Humidity and Wavelength Dependence of the Refractive Index of Hygroscopic Inorganic Aerosol Particles" by the authors Cotterell et al. submitted to Atmos. Chem. Phys. Discuss. This paper provides the refractive indices for aerosol particles composed of salt and water using several atmospherically relevant salts as a function of relative humidity from 100% RH to the efflorescence of the salt. Several wavelengths in the visible were used and dispersion coefficients were derived to obtain the refractive index at intermediate values. The significance of the article is summarized in section 4 "these ...[data] represent the most comprehensive description of refractive index for atmospherically relevant inorganic aerosol, fully characterizing the RI variation with both visible wavelength[s] and RH". I have only a few minor comments that should be addressed.*

*The introduction is very complete and gives good motivation for the current study.*

*How does phase function from this system relate to phase function measured from a polar nephelometer? Could raw data be shown?*

> **Response**
> We thank the reviewer for their positive comments on the significance of our refractive index parameterisations and the quality of the introduction. The phase function for both instruments is a measure of the variation in light scattering intensity with scattering angle although different angular ranges and resolutions, beam illumination geometries, collection optics and wavelengths make any direct comparison challenging. Here the angular scattering (phase function) is recorded with a high numerical aperture objective over an angular range (~ 50 degrees). The phase function is specifically used to show that our current phase function collection and data processing gives refractive indices that are consistent with those measured by Tang *et al*. (see Figure 3 for sodium chloride). Moreover, we show that there is excellent agreement between our recorded phase functions and the best-fit predictions from Mie theory (see Figure 2(b)). We also show that accurate and precise refractive index retrievals through fitting phase functions to Mie theory can be made only when optical illumination is provided by a source that is representative of the plane wave illumination assumed by Mie theory.
>
> For examples of raw phase function images, we now include in Section 3.1 of the current manuscript:
> 'We refer the reader to Figure 1 in (Cotterell et al., 2015a) and Figure 4 in (Carruthers et al., 2012) for examples of raw phase function images recorded using our instrumentation.'

*Why are the authors using one apparatus for one RH range and another for the higher RH range?*

> **Response**
> We assume that the reviewer is referring to Figure 3 of our manuscript. The measurements shown in Figure 3 serve to highlight the agreement between refractive index retrievals from our phase function instrumentation (and associated data processing routines) with refractive indices retrieved using other single particles methods, specifically with optically-tweezed droplets and light scattering measurements performed by Tang *et al*. using an electrodynamic balance. There is no particular reason why the RH

ranges do not perfectly match and neither must they: our sole aim is to ensure consistency in values of refractive index retrieved. Please note, the legend in Figure 3 has been corrected, where the optical tweezers and phase function retrieved refractive indices were mislabelled.

The following text has been added to the manuscript to section 3.2 (on page 10, line 15 onwards):
"Figure 3 shows $n_{633}$ for an aqueous sodium chloride particle retrieved from AOT measurements and the average of 3 measurements from PFs measured on the 405-nm SP-CRDS instrument, with good agreement between the techniques. Furthermore, $n_{633}$ from both techniques are consistent with the parameterisation of $n_{633}$ from Tang et al., within the RH uncertainty indicated corresponding to $\pm$ 2 % for AOT and PF measurements. Although the RH ranges of the new measurements do not perfectly match for the limited number of measurements compared here, the consistency between the new measurements and the previous parameterisation provided by Tang et al. is sufficient for validating the measurement approaches."

*I struggled at times to read this article due to the many, many acronyms. For example, in other papers on optical properties, "BB" means broadband, and here it means Bessel beam, which leads to confusion. Please cut down on their use.*

**Response**

We have reduced the use of acronyms as much as possible. We refer to our Bessel beam (BB) optical trap numerous (34) times in our manuscript, so we are inclined to keep the 'BB' acronym. However, to avoid confusion with 'broadband', we have now removed the several acronyms for ensemble broadband cavity enhanced spectroscopy (E-BBCES) on page 2, lines 5 – 15, and used the full expression instead.

*Fig 5: It is unclear why results vary so much from one another and are different than Tang et al. even for measurement at close to the same wavelength. I would expect less dispersion at larger wavelengths, but the differences look comparable in the 600nm region to the data for 405 vs 473 nm even though they are only separated by 13 nm.*

**Response**

The reviewer is correct in stating that dispersion should be less at larger wavelengths. Indeed, this is what we observe in our refractive index retrievals. This is somewhat difficult to infer from Figure 5 as the refractive index is plotted in the relative humidity domain. However, Figure 8 reports the same data from Figure 5 in the wavelength domain. From the changes in the slope of the refractive index dependence on wavelength, there is the general trend for all inorganic solutes of interest that dispersion is less strong at longer wavelengths. There are some refractive index retrievals which deviate from this trend; for example, the 633 nm RI is larger than the 589 nm RI for $NH_4HSO_4$ in Figure 8(d). We have already provided a description and account of these discrepancies in the manuscript. For example, see the last four paragraphs of Section 3.4 and the third-from-last paragraph of Section 4 which associate these discrepancies to calibration errors in RH probes and lack of both literature and experimental refractive index values when the RH is low and approaches the efflorescence RH of the inorganic salt.

*Fig 6: Why is a large deviation seen for ammonium bisulfate at low RH between values obtained with CRD and those obtained from the phase function?*

**Response**

We have discussed this observation in paragraph 3 of Section 3.5. The purpose of this Figure and associated discussion is to show that refractive index retrievals from phase functions cannot be performed reliably when the illuminating light field deviates strongly from the plane wave assumed by Mie theory. We show comparisons of retrieved refractive index values from phase functions associated with particle illumination by a focussed Bessel laser beam with values from cavity ring-down spectroscopy, with the cavity ring-down spectroscopy values shown to be both accurate and precise in our previous work. For all inorganic solute cases (not just for ammonium bisulfate), large deviations in the refractive index from Bessel beam illumination phase functions are observed from the expected (cavity ring-down spectroscopy) values.

We have added a sentence to the beginning of paragraph 3 of Section 3.5 to explicitly state, 'We now provide further evidence that reliable refractive index retrievals from fitting Mie theory to phase functions cannot be performed when the illuminating light field is a focussed Bessel beam.'

*How easily expandable are the techniques used in this paper to the study of mixed organic/inorganic mixtures?*

**Response**

We have performed preliminary experiments in our laboratory (unpublished) to study phase separation relative humidity for a mixed organic/inorganic system. Here, we measured the changes in phase functions and cavity ring-down time for single particles that existed as homogenous droplets at high RH but phase separated as the RH was lowered. The reviewer is certainly correct to highlight the study of complex mixtures as an interesting and important area of aerosol science, to which the application of our single-particle cavity ring-down spectrometer could make important contributions.